# Noise-induced quantum synchronization with entangled oscillations

Ziyu Tao [1,2,3,4,8], Finn Schmolke[5,8], Chang-Kang Hu[1,2,3,4,8], Wenhui Huang[2,3,4], Yuxuan Zhou[1,2,3,4], Jiawei Zhang[2,3,4], Ji Chu[1,2,3,4], Libo Zhang[2,3,4], Xuandong Sun[2,3,4], Zechen Guo[2,3,4], Jingjing Niu [1], Wenle Weng [6], Song Liu [1,2,3,4,7], Youpeng Zhong [1,2,3,4,7], Dian Tan [1,2,3,4,7] ✉, Dapeng Yu [1,2,3,4,7] ✉ & Eric Lutz[5] ✉

Random fluctuations can lead to cooperative effects in complex systems. We here report the observation of noise-induced quantum synchronization in a chain of superconducting transmon qubits with nearest-neighbor interactions. The application of Gaussian white noise to a single site leads to synchronous oscillations in the entire chain. We show that the two synchronized end qubits are entangled, with nonzero concurrence, and that they belong to a class of generalized Bell states known as maximally entangled mixed states, whose entanglement cannot be increased by any global unitary. We further demonstrate the stability against frequency detuning of both synchronization and entanglement by determining the corresponding generalized Arnold tongue diagrams. Our results highlight the constructive influence of noise in a quantum many-body system, and initiate the exploration of collective synchronization effects with stronger than classical correlations.

Noise is widely regarded as a nuisance that limits the transmission and processing of information[1]. The adverse effect of random fluctuations is even more dramatic in quantum physics, since quantum coherence and quantum correlations, two essential resources of quantum technologies[2], are highly susceptible to external disturbances[3]. Surprisingly, the nontrivial interplay between noise and nonlinear dynamics may induce order and organization[4–7]. Classical noise-induced phenomena, such as pattern formation, noise-induced transport and stochastic resonance, occur in a great variety of contexts, from physics and chemistry to biology and engineering[4–7]. However, observing the constructive role of noise in quantum systems is far more challenging[8–13].

Noise-induced synchronization is another counterintuitive consequence of random fluctuations[14–23]; it is, for instance, thought to be relevant for collective neuronal synchronization in the brain[21–23]. Synchronization is a general concept in classical[24–30] and quantum[31–40] physics: synchronous motion usually arises when coupled nonlinear oscillators adjust their internal rhythms, and oscillate in unison[24–40]. Synchronization phenomena have lately found interesting applications in communication systems[41,42] and in complex networks[43]. In noise-induced synchronization, collective oscillations arise through the constructive influence of a noise source. This classical effect has been observed in lasers[19,20] and in sensory neurons[21]. In the quantum regime, noise-induced synchronization has been predicted to occur in many-body systems and to lead to entangled collective oscillations[40]. So far, synchronous oscillations with stronger than classical, long-distance correlations have not been observed.

[1]International Quantum Academy, Futian District, Shenzhen, Guangdong, China. [2]Shenzhen Institute for Quantum Science and Engineering and Department of Physics, Southern University of Science and Technology, Shenzhen, China. [3]Guangdong Provincial Key Laboratory of Quantum Science and Engineering, Southern University of Science and Technology, Shenzhen, China. [4]Shenzhen Key Laboratory of Quantum Science and Engineering, Southern University of Science and Technology, Shenzhen, China. [5]Institute for Theoretical Physics I, University of Stuttgart, Stuttgart, Germany. [6]Institute for Photonics and Advanced Sensing (IPAS) and School of Physics, Chemistry and Earth Sciences, The University of Adelaide, Adelaide, SA, Australia. [7]Shenzhen Branch, Hefei National Laboratory, Shenzhen 518048, China. [8]These authors contributed equally: Ziyu Tao, Finn Schmolke, Chang-Kang Hu ✉ e-mail: tand@sustech.edu.cn; yudp@sustech.edu.cn; eric.lutz@itp1.uni-stuttgart.de

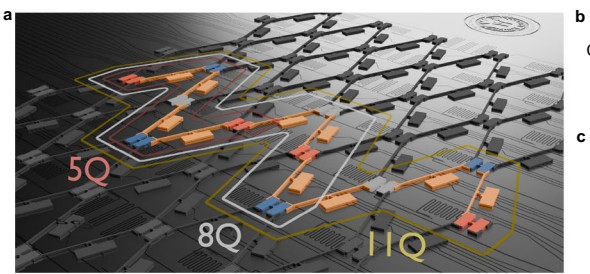

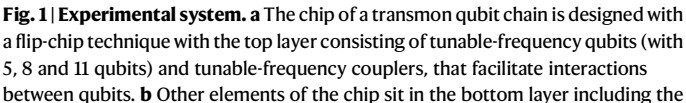

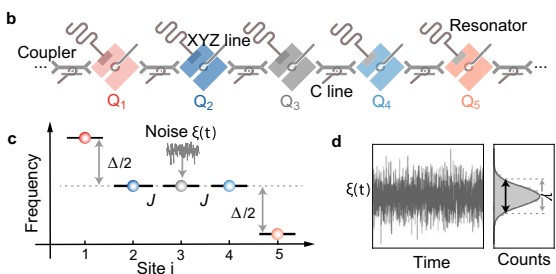

**Fig. 1 | Experimental system. a** The chip of a transmon qubit chain is designed with a flip-chip technique with the top layer consisting of tunable-frequency qubits (with 5, 8 and 11 qubits) and tunable-frequency couplers, that facilitate interactions between qubits. **b** Other elements of the chip sit in the bottom layer including the resonators, the qubit control and flux lines (*XYZ* lines) and neighboring coupler flux lines (*C* lines). **c** A one-dimensional *XY* chain of five transmon qubits, with coupling constant *J* and edge-spin frequencies detuned by Δ, is subjected to **d** Gaussian white noise ξ(t) with effective amplitude γ.

We here report the experimental realization of noise-induced quantum synchronization in a linear chain of superconducting transmon qubits with nearest-neighbor interactions[44]. We observe the occurrence of stable, synchronized oscillations of the magnetizations of the edge qubits when Gaussian noise is applied to a single qubit. We further show that the corresponding synchronized state is not only entangled, with nonzero concurrence[45], but that it is given by a maximally entangled mixed state which exhibits the maximum obtainable amount of entanglement for a given degree of mixedness[46–51]. Such states are regarded as direct generalizations of maximally entangled pure Bell states[46–51], and play an important role in mixed-state quantum information processing[52–55]. We additionally confirm the robustness of the observed quantum synchronization phenomenon to detuning of the natural frequencies of the qubits. We concretely obtain Arnold-tongue-like patterns[24–30], for both synchronization and entanglement, as a function of detuning and noise strength.

## Results

### Experimental system

Our experiments are implemented on a superconducting quantum processor, comprising a one-dimensional array of tunably coupled transmon qubits[44] (Fig. 1a, b). The qubits act as artificial spins, where the *j*th qubit frequency $\omega_j/(2\pi)$ can be controlled, in the range from ~3.2 GHz to ~4.6 GHz, by applying an external flux through the dedicated *Z* line of the corresponding qubit and coupler[56]. Likewise, the coupling constants $(J_j = J)/(2\pi)$ between the qubits is set to ~10 MHz by applying an external flux on the associated *C* line[56] (Supplementary Information). Each qubit can be individually addressed and driven into the excited state by applying a microwave pulse through its *XY* control line. The lattice model of the experiments can be described by the Hamiltonian of a one-dimensional quantum *XY* chain of *N* spins in a transverse field[57]

$$H_0 = \frac{\hbar J}{2} \sum_{j=1}^{N-1} \left( \sigma_j^x \sigma_{j+1}^x + \sigma_j^y \sigma_{j+1}^y \right) + \sum_{j=1}^{N} \hbar \omega_j \sigma_j^z, \quad (1)$$

where $\sigma_j^{x,y,z}$ are the local Pauli operators acting on site *j*. We apply Gaussian white noise ξ(t) with zero mean and autocorrelation $\langle \xi(t)\xi(t') \rangle = \Gamma\delta(t - t')$, with noise strength $\Gamma$, by locally modulating the natural frequencies of the individual qubits on the desired sites[58] (Supplementary Information) (Fig. 1c, d). This stochastic contribution corresponds to the addition of the Hermitian operator $\xi(t)\sigma_u^z$ (acting locally on site *u*) to the Hamiltonian (1). Stable synchronization of the entire quantum chain is predicted to occur, for a homogeneous chain with constant frequencies, $\omega_j = \omega$, and noise applied to a single site *u*, when the two conditions, $N = 5 + 3m$ and $u = 3n$ ($m, n \in \mathbb{N}$), are satisfied[40]. This leads to a decoherence-free subspace, that is decoupled from the surroundings[59], with only a single eigenmode, whose frequency determines the synchronization frequency. In the

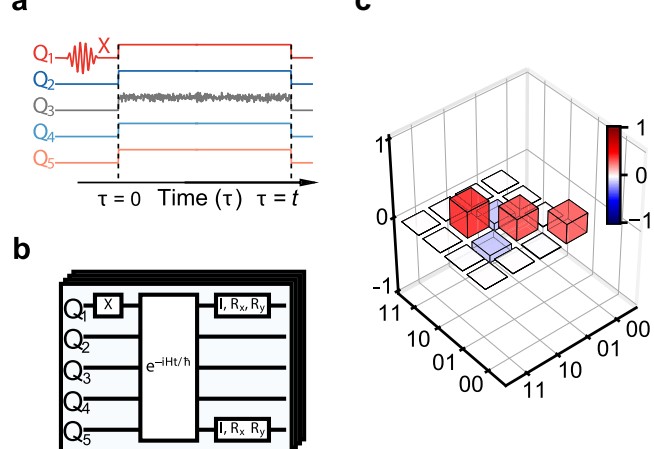

**Fig. 2 | State preparation. a** The system is initially prepared in a separable state where qubit $Q_1$ is prepared in the excited state while all the other qubits $Q_j$ of the quantum chain are in the ground state; external noise ξ(t) is applied to the central spin $Q_3$. **b** Quantum circuits used in the tomographic measurement of the two-qubit density matrix $\rho_{15}$ of the two end spins. **c** The measured real part of the density matrix $\rho_{15}$ of the synchronized edge qubits at time $Jt = 3\pi$ corresponds to a maximally entangled mixed state.

following, we analyze the occurrence of stable noise-induced quantum synchronization in a chain of *N* = 5 qubits (corresponding to *m* = 0) when the noise acts on the third qubit (corresponding to *n* = 1) in the middle of the chain (Fig. 1c). This is the smallest spin chain in which noise-induced quantum synchronization is expected to appear. Additional cases with larger number of spins (*N* = 8 and 11), as well as with violations of the synchronization conditions are presented in the Supplementary Information.

### Noise-induced quantum synchronization

We initially prepare the quantum spin chain in the separable state $|\Psi(0)\rangle = |1\rangle \otimes |0\rangle^{\otimes 4}$, where $|1\rangle$ and $|0\rangle$ denote the respective excited and ground states of the qubits (Fig. 2a, b). The state of each qubit is determined by measuring the state-dependent transmission of a dedicated readout resonator, with frequency centered around 6.1 GHz, coupled to each qubit using a dispersive readout scheme (Supplementary Information).

Figure 3 displays the temporal evolution of the measured local *z*-polarization $\langle \sigma_j^z \rangle$ of the individual qubits, without (Fig. 3a) and with (Fig. 3b) Gaussian noise applied to the middle of the chain (the reduced noise strength is γ = Γ/J ≈ 1.3). In the absence of noise, the initial excitation travels through the chain in a wave-like manner, is reflected at the open boundaries, and bounces back and forth between the edges

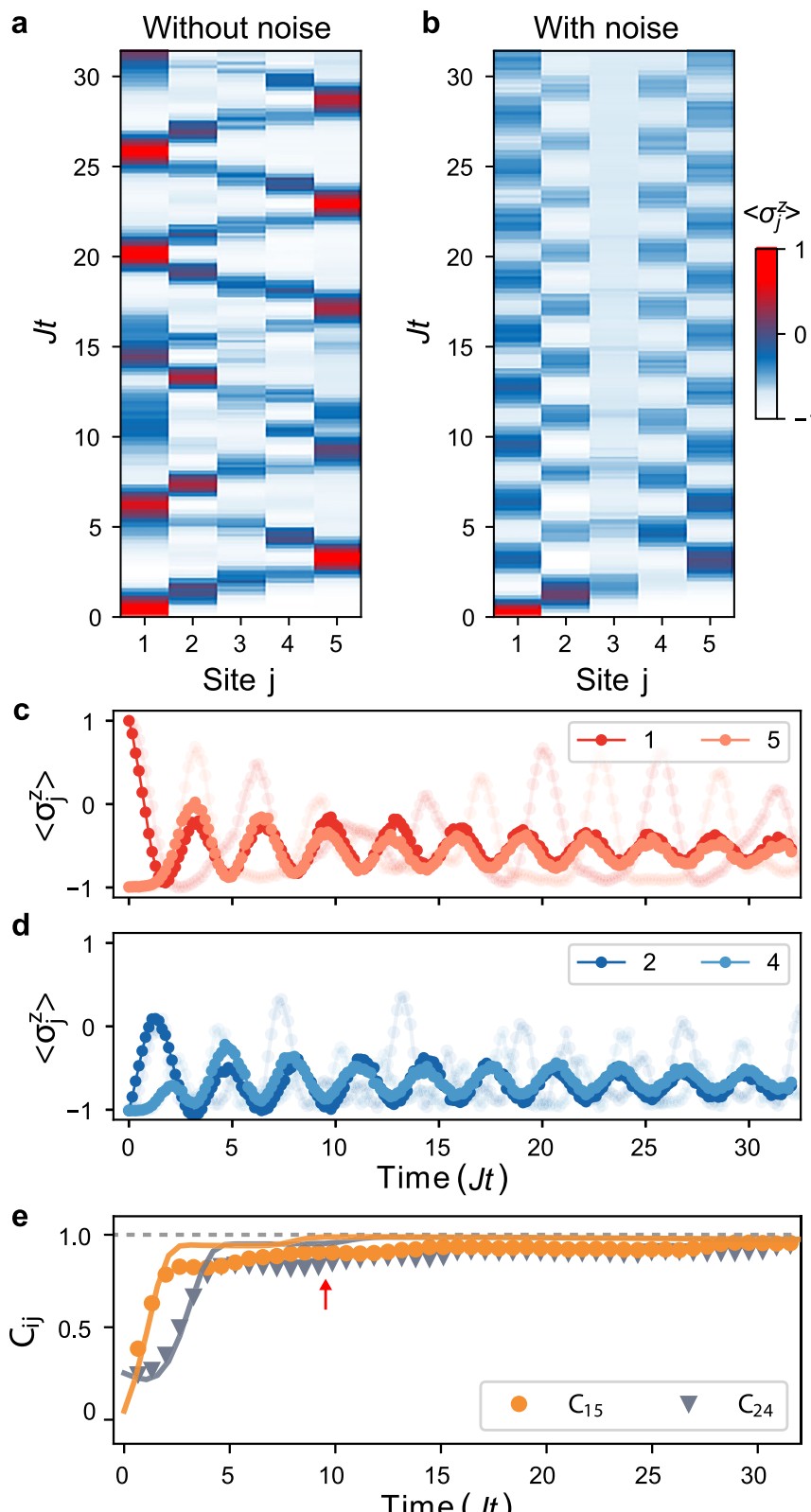

**Fig. 3 | Noise-induced quantum synchronization. a** The local $z$-polarizations $\langle\sigma_j^z\rangle$ of a chain of five qubits do not synchronize during noise-free unitary dynamics. **b** Synchronized oscillations, with frequency $\sim 2J = 20$ MHz, between **c** qubits 1–5 and **d** qubits 2–4 occur, when Gaussian white noise with strength $\gamma = 1.3$ is applied to the third spin. The associated unsynchronized unitary evolution is shown in the background. **e** Pearson correlation coefficients $C_{15}$ and $C_{24}$ of qubits 1–5 and qubits 2–4 (symbols) converge to one, indicating perfect correlation between the corresponding qubits. The synchronized regime, $C_{ij} \geq 0.9$, is attained for $Jt \geq 3\pi$ (red arrow). Good agreement with theory (solid lines) is obtained.

of the chain, without collective coordination between the individual qubits (Fig. 3a). By contrast, in the presence of Gaussian noise, qubits 1 and 5 (as well as qubits 2 and 4) oscillate in phase at a single frequency of ~20 MHz, in agreement with the predicted value $2J$, after some transient (Fig. 3b, c). The synchronization frequency is set by the coupling constant of the quantum chain and not by the eigenfrequencies of the qubits. The faint lines in the background of Fig. 3c, d represent the measured unsynchronized, quasi-unitary dynamics of the magnetizations, when the external Gaussian noise is switched off.

In order to quantitatively characterize the synchronized oscillations of the respective polarizations, we use the Pearson correlation coefficient, a standard measure of quantum synchronization[38–40], defined as the ratio of the covariance and the respective standard deviations, $C_{ij} = \text{cov}\left(\langle\sigma_i^z\rangle, \langle\sigma_j^z\rangle\right)/\sqrt{\text{var}(\langle\sigma_i^z\rangle)\text{var}(\langle\sigma_j^z\rangle)}$[60]. The latter quantity provides a measure of the degree of linear correlation between observables; it ranges from −1 (corresponding to anticorrelated oscillations) to +1 (indicating correlated evolution). Figure 3e shows the Pearson correlation coefficients $C_{15}$ and $C_{24}$ of the measured z-polarizations of qubits 1–5 and qubits 2–4, as a function of time (symbols). Both quickly converge to one, demonstrating almost perfectly correlated oscillations over the entire duration of the experiment, in good agreement with the theoretical simulations (solid lines) (Supplementary Information). We specifically consider two qubits to be synchronized for Pearson coefficients $C_{ij} \geq 0.9$, which happens for times $Jt \geq 3\pi$ (red arrow). We also mention that synchronized oscillations occur for arbitrary initial states as long as they have a nonzero overlap with the synchronized mode (Supplementary Information).

The synchronization condition ensures that the noise dynamically suppresses all the eigenmodes of the system except one, as seen in the measured Fourier spectrum of the magnetizations (Fig. 4a, b). The frequency ~20 MHz of the surviving mode determines the synchronization frequency (see below). This synchronization effect may thus be regarded as a quantum generalization of the classical synchronization mechanism known as "suppression of natural dynamics"[28–30]. It is worth mentioning that dissipation is necessary to induce stable synchronization in a quantum many-body system, since the required permutation invariance of local observables, such as $\sigma_j^z$, is not guaranteed for purely unitary evolution[39] (Supplementary Information).

We next analyze the robustness of the synchronous oscillations. To that end, we introduce a variable detuning $\Delta$ between the natural frequencies of the synchronized end qubits via a term $H_1 = (\hbar\Delta/2)(\sigma_1^z - \sigma_5^z)$ added to the system Hamiltonian (1) (Fig. 1c). Figure 5a displays the measured Pearson correlation coefficient $C_{15}$ at the onset of the synchronization regime $Jt = 3\pi$, when both the reduced noise amplitude $\gamma$ and the detuning $\Delta$ are varied. We recognize a structure which is reminiscent of an Arnold tongue which defines the synchronized domain of classical synchronization phenomena[24–30]. Noise-induced quantum synchronization appears to be a robust effect that occurs in a wide range of parameters of the system. We note that the synchronization region is enlarged when the detuning is reduced and when the noise strength is increased; it is thus easier to synchronize identical spins with strong noise

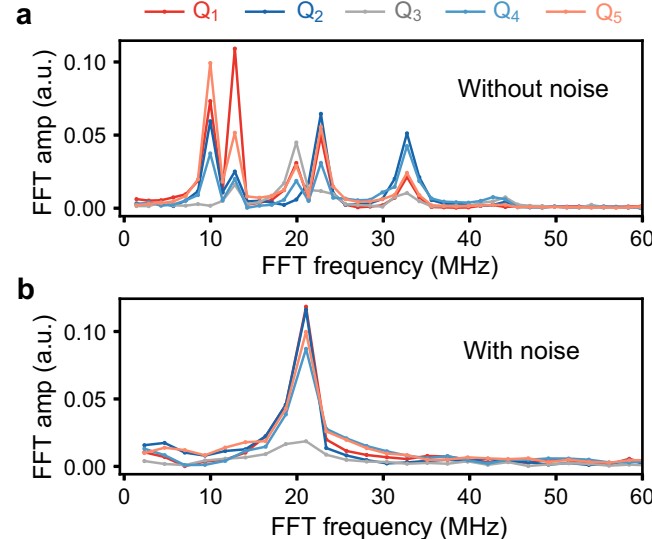

**Fig. 4 | Eigenmode suppression. a** Modulus of the Fourier transform of the magnetizations after being synchronized for time $Jt = 3\pi$ without, and **b** with noise, showing the suppression of all the system eigenmodes, except one.

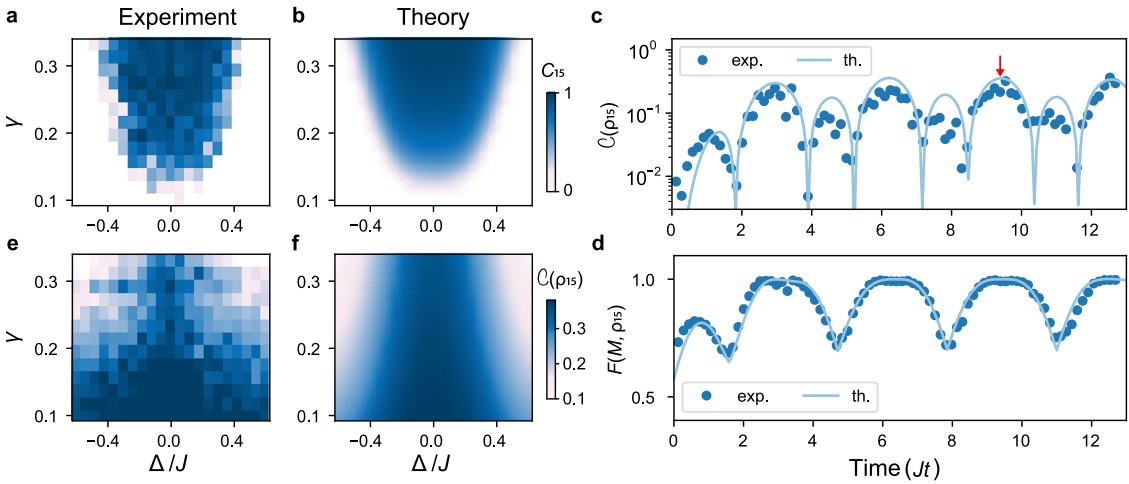

**Fig. 5 | Stability regions and entanglement.** Arnold tongue of synchronization. Experimental data (**a**) and numerical simulation (**b**) for the Pearson correlation coefficients $C_{15}$, extracted at a given time $Jt = 3\pi$ (red arrow), as a function of noise amplitude $\gamma$ and detuning $\Delta$ between the two end spins. Larger values of detuning are detrimental, but the system can still be synchronized by increasing the noise strength. Concurrence $\mathcal{C}(\rho_{15})$ (**c**) and fidelity $F(M, \rho_{15})$ (**d**) of the predicted maximally entangled mixed state ($M$) and the measured two-qubit density operator ($\rho_{15}$) as a

function of time. Both quantities display non-vanishing steady oscillations for $Jt \geq 2\pi$, showing that the synchronized two-qubit state is entangled Good agreement between data (dots) and theory (lines) is observed. Entanglement tongue. Experimental data (**e**) and numerical simulation (**f**) for the concurrence $\mathcal{C}(\rho_{15})$ at a given time $Jt = 3\pi$ as a function of noise amplitude $\gamma$ and the detuning $\Delta$. Increasing both the detuning and the noise strength diminishes the amount of entanglement in the system.

(provided the noise amplitude remains below the noise-induced quantum Zeno regime[61,62]). We again obtain good agreement with theoretical simulations (Fig. 5b).

## Maximally entangled mixed states

Entanglement is a fundamental resource in quantum information science; mechanisms creating entangled states are hence of great importance[2]. There seems, however, to be no direct relationship between quantum synchronization and quantum correlations in general[63–65]. In view of the detrimental influence of noise on quantum properties[3], it is therefore all the more remarkable that noise-induced synchronization is expected to give rise to entangled synchronized edge qubits[40]. In order to test this feature, we tomographically reconstruct the two-qubit state of the two end spins[66] as illustrated in Fig. 2c and evaluate the concurrence, $\mathcal{C}(\rho_{15}) = \max(0, \sqrt{\kappa_1} - \sqrt{\kappa_2} - \sqrt{\kappa_3} - \sqrt{\kappa_3})$; the operator $\rho_{15}$ is here the reduced density matrix of the two edge qubits and $\kappa_n$ are the ordered eigenvalues of the product $\rho_{15}\widetilde{\rho}_{15}$, with $\widetilde{\rho}_{15}$ the spin flipped state[45]. Figure 5c shows that $\mathcal{C}(\rho_{15})$ exhibits nonzero steady oscillations, clearly indicating the presence of synchronized entangled edge qubits. This observation reveals an intriguing connection between collective quantum behavior and nonclassical correlations. The concurrence reaches a steady state for $Jt \geq 2\pi$, thus slightly before the two edge qubits are fully synchronized ($Jt \geq 3\pi$).

The reconstructed two-qubit state $\rho_{15}$ at time $Jt = 3\pi$ is explicitly given in Fig. 2c: it has the form of a maximally entangled mixed state which can be parametrized as $\rho = p_1|\Psi^-\rangle\langle\Psi^-| + p_2|00\rangle\langle00| + p_3|\Psi^+\rangle\langle\Psi^+| + p_4|11\rangle\langle11|$ with $p_1 \geq p_2 \geq p_3 \geq p_4$[46], where $|\Psi^-\rangle = (|01\rangle - |10\rangle)/\sqrt{2}$ and $|\Psi^+\rangle = (|01\rangle + |10\rangle)/\sqrt{2}$ are the usual Bell states[2] and $\sum_k p_k = 1$. The concurrence of the maximally entangled mixed state can be analytically determined as $\mathcal{C} = p_1 - p_3 - 2\sqrt{p_2 p_4}$[46]. The fidelity of the measured state $\rho_{15}$ and the theoretical maximally entangled mixed state $M$ given, for the parameters of the experiment, by

$$M = \begin{pmatrix} 0 & 0 & 0 & 0 \\ 0 & 1/3 & -1/6 & 0 \\ 0 & -1/6 & 1/3 & 0 \\ 0 & 0 & 0 & 1/3 \end{pmatrix}, \qquad (2)$$

is $F(M, \rho_{15}) = \mathrm{Tr}\left[\sqrt{\sqrt{M}\rho_{15}\sqrt{M}}\right] = 99.3\%$ (Supplementary Information). The maximally entangled mixed state is hence already created at the beginning of the synchronized regime (the fidelity between measured and theoretical states at $Jt = 4\pi$ is $F = 99.6\%$). The time evolution of the fidelity of the measured two-qubit state and the theoretical maximally entangled mixed state is shown in Fig. 5d. It exhibits a behavior similar to that of the concurrence (Fig. 5c). In particular, it displays steady oscillations for $Jt \geq 2\pi$. Maximally entangled mixed states define a class of quantum states for which no more entanglement can be created by any global unitary operations[46–51]. They have the interesting property that they are more entangled than Werner states with the same purity[46–51]. So far, maximally entangled mixed states have only been generated in optical systems[49–51].

Like the Pearson correlation coefficient $C_{15}$, the concurrence $\mathcal{C}(\rho_{15})$ is robust to detuning of the edge qubit frequencies. It appears in a large parameter domain (Fig. 5e), and exhibits an (inverted) Arnold-tongue-like structure that results from the competition of two different mechanisms: on the one hand, entanglement between the initially separable end spins is created through the unitary evolution of the system[67–70]; on the other hand, noise, which drives the quantum synchronization process, destroys quantum correlations. The value of the effective noise strength $\gamma$, which controls the synchronization time[40], sets the maximal amount of entanglement that the synchronized state can have once it has reached the (quasi)-stationary state in the decoherence-free subspace. The concurrence of the maximally entangled mixed state thus decreases when the noise amplitude or the

detuning are increased. The entanglement tongue seen in Fig. 5e is often regarded as a quantum generalization of the classical Arnold tongue[71]. As before, good agreement with theoretical simulations is found (Fig. 5f).

## Discussion

Classical synchronization gives rise to fascinating collective oscillation phenomena[24–30]. On the other hand, entanglement has been recognized as a powerful resource for quantum applications[2]. Both are ubiquitous in current science and technology. We have taken a first step towards merging these two fields by demonstrating the occurrence of entangled, quantum synchronization in a chain of transmon qubits. Applying Gaussian white noise to one site of the chain, perfectly correlated in-phase oscillations at a frequency set by the coupling constant of the chain, with a Pearson coefficient close to one, have been observed. These findings provide a unique illustration of the nontrivial interplay between noise and unitary dynamics in a quantum many-body system, leading to collective behavior, and at the same time, to the creation of distant quantum correlations with nonzero concurrence. In view of their generality, we expect these results to be important for future studies of quantum-enhanced synchronization, including synchronization-based quantum communication[72,73], complex quantum networks[74,75] and quantum metrology[76].

## Data availability

The datasets generated during and/or analyzed during the current study are available from the corresponding authors on reasonable request.

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

## Acknowledgements

This work was supported by the National Natural Science Foundation of China (11934010, 12205137, 12004167), the Key-Area Research and Development Program of Guangdong Province (Grants No. 2018B030326001 and 2020B0303030001), the ChinaPostdoctoral Science Foundation (Grants No. 2020M671861 and 2021T140648), the Guangdong Provincial Key Laboratory (Grant No. 2019B121203002), Technology and Innovation Commission of Shenzhen Municipality (KQTD20210811090049034), the Innovation Program for Quantum Science and Technology (Grants No. 2021ZD0301703). F.S. and E.L. acknowledge financial support from the Vector Foundation and the DFG (Grant FOR 2724).

## Author contributions

D.T., D.Y. and E.L. supervised the project. D.T. initiated the project and designed the experiment. Z.T. conducted the measurements with assistance from C.-K.H., W.H, J.Z., J.C., X.S., Z.G., J.N, and Y.Zhong. F.S. developed the theory and did the theoretical simulations supervised by E.L. Y.Z. and L.Z. fabricated the devices supervised by S.L. Z.T., F.S., C.-K.H., W.W., D.T. and E.L. analyzed the data. Z.T., F.S., D.T. and E.L. wrote the manuscript with feedback from all the authors.

## Funding

## Competing interests

The authors declare no competing interests.
