## [Transparent Peer Review file · Nature Communications]

Noise-induced quantum synchronization with entangled oscillations

Corresponding Author: Professor Eric Lutz

Version 0:

Reviewer comments:

Reviewer #1

(Remarks to the Author)

The authors of this manuscript present detailed measurements of different spin chains, subjected to noise at specific locations along the chain, and observe the predicted synchronization of the qubits. While the application of this in quantum communication or quantum computing is somewhat difficult to imagine, the physics observed is quite interesting, especially in the case of a random initial state, as shown in the supplement. Without that random initial state synchronization, most of the other initial states exhibit a lot of symmetry, and I would have suspected that dimerization is occurring, but I think I am convinced of the phenomenon they are claiming because of the random initial state. As a reader, I found the supplement very important for my understanding of the document.

However, while reading the manuscript, there are a few things that can be done to improve the manuscript:

(1) Figure 2 presents an FFT of the magnetization before showing what those traces look like (Fig. 3c,d). I find this odd and hard to follow.

(2) Some experimental details are lacking. For instance, the authors mention the ratio of EJ_1/EJ_2 but do not state what the individual E_j s are or E_c s. This is important information that should be shared. At the same time, they state that the readout resonators are all at 6.1 GHz; however, Supplemental Figure 8 shows that some multiplexing is occurring. It's important to report these numbers. Additionally, they state the readout fidelity for one of the qubits but not for the others. It is very important to share all relevant experimental details; otherwise, it makes it difficult to reproduce their results.

(3) The claim that Supplement Figure 16 shows evidence of synchronization is a bit of a reach. If you want to claim that, I would show the Correlation coefficients, but I am suspicious that there is a reason those have not been included. Again, transparency is key. If you are going to claim synchronization in this longer chain, back it up with additional data analysis. Along these lines, is dissipation the main reason why this synchronization degrades? How is this influenced by system size? I think it's essential to be transparent here and acknowledge that things tend to worsen as the system size increases.

In summary, my reading of this paper remains quite positive, and I believe there is an appetite for synchronization physics. I suggest publication after addressing all the concerns mentioned above and applaud the authors on their experimental achievement.

Response to Reviewer #1

We thank the Referee for the positive report.

“1) However; while reading the manuscript, there are a few things that can be done to improve the manuscript: (1) Figure 2 presents an FFT of the magnetization before showing what those traces look like (Fig. 3c,d). I find this odd and hard to follow.”

As suggested, we have split Fig. 2 into two: we now present the old Figs. 2d,e with the FFT of the magnetizations in the new Figs. 4a,b (after the traces shown in Figs. 3c,d) to avoid any confusion. We have also moved the description of that figure from page 2 to page 3 of the main text.

“2) Some experimental details are lacking. For instance, the authors mention the ratio of E_{J1}/E_{J2} but do not state what the individual E_J s are or E_C s. This is important information that should be shared. At the same time, they state that the readout resonators are all at 6.1 GHz; however, Supplemental Figure 8 shows that some multiplexing is occurring. It’s important to report these numbers. Additionally, they state the readout fidelity for one of the qubits but not for the others. It is very important to share all relevant experimental details; otherwise, it makes it difficult to reproduce their results.”

As requested, we have now added Table I on page 4 of the Supplementary Information listing the individual values of E_J and E_C for each qubit. Regarding the readout resonators, we have clarified in the main text that the resonators are *centered* around 6.1 GHz rather than being exactly the same. The specific resonator frequencies, readout fidelities for each qubit, and other relevant experimental parameters are also given in the new Table I for completeness and reproducibility (the table is reproduced on the next page for convenience).

“3) The claim that Supplement Figure 16 shows evidence of synchronization is a bit of a reach. If you want to claim that, I would show the Correlation coefficients, but I am suspicious that there is a reason those have not been included. Again, transparency is key. If you are going to claim synchronization in this longer chain, back it up with additional data analysis. Along these lines, is dissipation the main reason why this synchronization degrades? How is this influenced by system size? I think it’s essential to be transparent here and acknowledge that things tend to worsen as the system size increases.”

As recommended, we have included the correlation coefficients for the old Fig. S16 (now Fig. S12 in the separated Supplementary Information) for the 11-spin chain in the new Fig. S13 (see Fig. 1 below). We did not try to hide the corresponding analysis. Indeed, even in this case, the Pearson correlation coefficients show the occurrence of synchronization. However, the Referee is correct: synchronization becomes experimentally more challenging as the system size increases. We now indicate in Sec. “V. Synchronized oscillations for longer spin chains” of the Supplementary Information that this degradation arises from several factors: increased dissipation through the qubits and couplers, nonuniformities in qubit frequencies and coupling strengths, as well as by transitions to higher energy levels of the transmons (which thus start to slightly deviate from ideal two-level systems). For completeness, we have also included the Pearson correlation coefficient for the random initial state in the new Fig. S16 (see Fig. 2 below).

Figure 1: Pearson correlation coefficients of the $N = 11$ qubit chain for the single excitation initial state shown in Fig. S12. Synchronization becomes more challenging for larger chains due to dissipation, nonuniform couplings and excitation of higher energy levels.

Qubit index	f_{10}^{idle} (GHz)	E_C (GHz)	E_{J1} (GHz)	E_{J2} (GHz)	f_{rr} (GHz)	F_0 (%)	F_1 (%)	T_1 (μs)	T_{2r} (μs)
1	4.420	0.239	9.71	2.99	6.002	94.9	93.6	36.10	3.3
2	4.559	0.215	10.05	3.26	6.287	93.3	91.4	52.60	6.8
3	4.402	0.221	9.94	2.93	6.024	98.2	92.7	47.90	4.7
4	4.595	0.212	10.45	3.20	6.232	95.3	93.6	52.10	17.6
5	4.433	0.221	9.78	3.08	6.058	96.3	92.2	79.90	6.0
6	4.665	0.216	10.85	3.10	6.166	94.3	86.4	34.60	5.2
7	4.396	0.221	10.25	2.93	6.140	96.8	94.6	74.00	4.4
8	4.527	0.216	10.95	2.09	6.098	97.1	91.9	59.40	22.9
9	4.371	0.224	9.80	3.02	6.197	97.0	94.7	58.10	4.6
10	4.601	0.213	10.19	3.46	6.058	94.3	89.7	21.60	10.8
11	4.491	0.222	9.77	3.05	6.255	96.8	94.8	50.70	10.1

Table 1: Device parameters. This table lists the parameters of 11-qubit 1D chain on a superconducting quantum processor including qubit frequencies, charging energy (E_C), Josephson energy (E_j), frequencies of readout cavities, readout fidelity and the coherence times for each qubit.

Figure 2: Pearson correlation coefficients of the $N = 5$ qubit chain for the random initial state shown in Fig. S15.